# Rehabilitation of Difficult-to-Wean, Tracheostomized Patients Admitted to Specialized Unit: Retrospective Analyses over 10-Years

**DOI:** 10.3390/ijerph19052982

**Published:** 2022-03-03

**Authors:** Stefania Costi, Antonio Brogneri, Chiara Bagni, Giulia Pennacchi, Claudio Beneventi, Luca Tabbì, Daniela Dell’Orso, Riccardo Fantini, Roberto Tonelli, Gianfranco Maria Beghi, Enrico Clini

**Affiliations:** 1Surgical, Medical and Dental Department of Morphological Sciences Related to Transplant, Oncology and Regenerative Medicine CHIMOMO, University of Modena and Reggio Emilia, 41121 Modena, Italy; stefania.costi@unimore.it; 2Physical Medicine and Rehabilitation Unit, Azienda Unità Sanitaria Locale-IRCCS di Reggio Emilia, 42123 Reggio Emilia, Italy; 3Respiratory Rehabilitation of Ospedale Villa Pineta-KOS Group, 41026 Pavullo nel Frignano (MO), Italy; antonio.brogneri@libero.it (A.B.); beneventi.claudio@villapineta.it (C.B.); dellorso.daniela@villapineta.it (D.D.); g.beghi@hotmail.it (G.M.B.); 4School of Physiotherapy, Surgical, Medical and Dental Department of Morphological Sciences Related to Transplant, Oncology and Regenerative Medicine CHIMOMO, University of Modena and Reggio Emilia, 41121 Modena, Italy; 257409@studenti.unimore.it (C.B.); 239944@studenti.unimore.it (G.P.); 5Respiratory Diseases Unit, Department of Medical and Surgical Sciences SMECHIMAI, University Hospital of Modena, University of Modena and Reggio Emilia, 41121 Modena, Italy; lucatabbi@gmail.com (L.T.); fantini.riccardo@yahoo.it (R.F.); roberto.tonelli@me.com (R.T.)

**Keywords:** tracheostomy, difficult weaning, rehabilitation

## Abstract

Rehabilitation outcomes of difficult-to-wean tracheostomized patients have been reported in relatively small case studies and described for a limited time span. This study describes the characteristics and clinical outcomes of a large cohort of tracheostomized patients admitted to a specialized weaning unit over 10 years. We retrospectively analyzed data collected from January 2010 to December 2019 on difficult-to-wean tracheostomized patients who underwent comprehensive rehabilitation. Clinical characteristics collected at admission were the level of comorbidity (by the Cumulative Illness Rating Scale—CIRS) and the clinical severity (by the Simplified Acute Physiology Score—SAPS II). The proportions of patients weaned, decannulated, and able to walk; the change in autonomy level according to the Bristol Activities of Daily Living (BADL) Scale; and the setting of hospital discharge was assessed and compared in a consecutive 5-year time period (2010–2014 and 2015–2019) subgroup analysis. A total of 180 patients were included in the analysis. Patient anthropometry and preadmission clinical management in acute care hospitals were similar across years, but the categories of underlying diagnosis changed (*p* < 0.001) (e.g., chronic obstructive pulmonary disease—COPD—decreased), while the level of comorbidities increased (*p* = 0.003). The decannulation rate was 45.6%. CIRS and SAPS II at admission were both significant predictors of clinical outcomes. The proportion of patients whose gain in BADL score increased ≥ 2 points decreased over time. This study confirms the importance of rehabilitation in weaning units for the severely disabled subset of tracheostomized patients. Comorbidities and severity at admission are significantly associated with rehabilitation outcomes at discharge.

## 1. Introduction

It is estimated that 13–20 million people worldwide annually require respiratory support in intensive care units (ICUs) due to a compromised cardiorespiratory system, which results in acute respiratory failure (ARF) [1]. Invasive mechanical ventilation (iMV) is the most common intervention in patients admitted to these units [2]. Numerous patient-related factors and the causes underlying ARF often make the weaning process from MV difficult [3,4]. These patients usually present with a compromised health status, which can last long after hospital discharge [5]. Specialized facilities have therefore been set up recently to manage prolonged or difficult weaning processes. These facilities are known to be more effective than traditional ICUs in achieving the multiple desired outcomes in this population, including better perceived health status [6].

In addition to those factors linked to the higher risk of prolonged ICU stay (e.g., age, chronic underlying diseases, prolonged period of MV during the acute phase), these patients, even more so if tracheostomized, usually experience muscle weakness with severe and long-lasting limitations in functional independence [7]. Therefore, the approach to prolonged weaning preferably consists of multidisciplinary programs that include intensive physiotherapy [3] in order to minimize bed-rest syndrome, reduce length of stay, and promote patients’ autonomy in activities of daily living (ADL) [8,9]. Nevertheless, this clinical approach is not widely implemented in ICU units due to organizational limitations and/or restricted resources. Moreover, rehabilitation outcomes of difficult-to-wean tracheostomized patients have been reported in relatively small case studies and described for a limited time span.

The aim of this retrospective study was to describe patient characteristics and clinical outcomes following comprehensive rehabilitation in a large cohort of tracheostomized patients admitted to a specialized weaning unit over 10 years. We also compared the outcomes achieved in two consecutive 5-year time periods (2010–2014 and 2015–2019) since these periods were representative of more homogeneous approaches delivered, and we examined associations between the patients’ characteristics and their outcomes.

## 2. Methods

### 2.1. Study Design and Data Sources

This study is a retrospective analysis of data collected on difficult-to-wean tracheostomized patients who were admitted to the Weaning Unit of Villa Pineta-Kos Care Hospital in Gaiato (Pavullo, Italy), a regional rehabilitation center, from January 2010 to December 2019. Data were collected from the hospital’s digital administration database. All patients admitted to the Weaning Unit among two subsequent time periods (namely 2010–2014 and 2015–2019) were considered eligible for the analysis. Patients with degenerative neuromuscular diseases, those for whom the outcome of weaning and decannulation cannot be predicted beforehand and those who were considered not compliant to the rehabilitation programs, were excluded a priori. Patients who died during hospital stay were further excluded; thus, only those who completed the rehabilitation program were included in the analysis plan. Details on patient selection are shown in Figure 1.

The study was approved by the local ethics committee (protocol number AOU: 0002755/21, date 27 January 2021). Specific informed consent to use the data for the purposes of this study was obtained from patients or their relatives.

### 2.2. Comprehensive Rehabilitation Program

The rehabilitation program for tracheostomized patients with prolonged weaning was carried out at the Weaning Unit of Villa Pineta-Kos Care Hospital, a semi-intensive care unit with eight monitored beds. A multidisciplinary dedicated team that included physicians, specialized nurses, physiotherapists, nutritionists, and psychologists was responsible for the intensive rehabilitation program (IRP) and worked together to achieve the goals of recovering the patients’ respiratory and neuromuscular autonomy. The nurses and physiotherapists to patients ratio was 1:4 for both professionals. 

In the first 24–48 h from admission, the team carried out a complete patient assessment, including potential risks factors and complications. Afterward, the team members met to define the individual rehabilitation program (IRP) and to assign the patient to a specific protocol based on his/her triage score. All patients received a 1 h treatment twice daily whose intensity was tuned according to clinical severity. IRP included intensive physiotherapy and consisted of peripheral and respiratory muscle reconditioning, airway secretion management, and respiratory/breathing exercises. Cough assistance was used in case of established neuromuscular impairment. Positive pressure ventilation was used with plugged trach for patients with chronic hypercapnia with the aim of facilitating adaptation. The goals of the IRP were to achieve weaning up to decannulation and to increase the patient’s Bristol Activities of Daily Living (BADL) score by improving peripheral muscle mass and strength. Over the course of the rehabilitation program, the team regularly reassessed the patient to verify progress and/or modify goals. 

### 2.3. Decannulation Protocol 

According to the local protocol, the patient was considered eligible for decannulation once the following criteria were met:Patient was alert, oriented, and responsive to commands;Patient was able to manage oral secretions without risk of aspiration;Patient was no longer dependent on a ventilator for assisted breathing;The need for tracheal suctioning was less than once per day;Patient had the tracheostomy tube downsized to a size 4 Shiley or similar tracheostomy tube, and no breathing difficulty in the presence of this tube was reported;Successful 12 h occlusion test of the downsized tracheostomy tube was performed.

Once the previous criteria were met, the trach was plugged for 24 h, and the patient was monitored for respiratory difficulty or suction requirement. If this test was successful, the trach tube was removed. The patient was placed supine, the tube was removed, the opening into the neck was covered with sterile gauze, and tape was placed over the gauze. The patient was instructed to occlude the gauze with the fingertip in case of cough or need for speech. The gauze and the tape were changed at least once per day (more often if needed) until the hole in the neck healed itself and closed over the following days. In case of hole persistency, the opening into the neck skin was surgically closed. The patient was further discharged 3 to 5 days following successful decannulation.

### 2.4. Data Collection and Outcome Assessment

Within 24–48 h from admission (T0), the patients’ general characteristics, underlying disease, level of comorbidity, and clinical severity according to the Cumulative Illness Rating Scale (CIRS) and the Simplified Acute Physiology Scale (SAPS II) scores, the number of days spent in an acute care hospital before admission to the Weaning Unit, and the number of days from tracheostomy were recorded. 

Once rehabilitation was completed (T1), the study outcomes—the proportion of patients weaned from invasive mechanical ventilation (YES/NO), decannulated (YES/NO), able to walk (YES/NO), changes in autonomy level according to the BADL score (0 or less = NO, any point increase = YES), and the setting of discharge (home/residential care facility/acute care hospital)—were recorded. 

### 2.5. Statistical Analysis

A descriptive analysis, including average, median, standard deviation, and minimum and maximum values, was carried out on the data collected. Comparisons of continuous variables were performed using *t*-test, whereas categorical variables were reported as numbers and percentages (%) and compared by χ^2^ test or Fisher’s exact test, as appropriate.

Ordinal or binomial logistic regression analysis to observe the correlation between patient characteristics at admission and clinical rehabilitation outcomes was also conducted when appropriate.

The statistical analyses were carried out using Jamovi 1.2.27 package; a *p*-value less than 0.05 was considered statistically significant.

## 3. Results

### 3.1. General Characteristics

A total of 320 individuals was considered eligible for analysis over the pre-specified time period. Of these, 140 individuals (44%) admitted to the Weaning Unit over 10 consecutive years (1 January 2010 to 31 December 2019) were excluded due to the predefined inclusion criteria. Of note, 22/140 patients (16%) were excluded because they were not able to participate in intensive physiotherapy. Therefore, 180 patients were included in the analysis (Figure 1); the number of patients included by year ranged from 15 (in 2015) up to 22 (in 2010). The patients’ general and anthropometric data, clinical diagnosis, and severity at T0 were recorded for the overall population and grouped according to the two consecutive 5-year time periods (Table 1, part 1). Overall, the patients’ average age was 73 ± 10.1 years. Males were slightly more represented (56.7%), with significant differences across the two time periods examined (*p* = 0.020). The underlying conditions at admission were reported in each patient, being respiratory diseases as the most represented. The CIRS and SAPS II mean scores were respectively 19 ± 3.9 points and 21.5 ± 14.1%, with variations over time for the former (*p* = 0.002). Overall, preadmission days in an acute care hospital (mean 36.4 ± 26.7 days) and number of days from tracheostomy (mean 22.1 ± 22.3 days) did not substantially change over time. 

### 3.2. Clinical Outcomes

The program’s outcomes (see Methods) achieved in the whole population and in the two periods are reported in Table 1, Part 2. The proportion of weaned individuals tended to be higher in the second period. Among those who did not achieve weaning, around one-fifth was partially weaned and discharged under nocturnal mechanical ventilation, with no differences between the two 5-year periods. Nonetheless, the proportion of patients who participated in comprehensive rehabilitation and who improved their BADL score decreased, in particular for the categories of individuals who achieved an improvement ≥ 2 points. Contrarily, the proportion of patients whose score remained unchanged increased. Altogether, 58.7% and 41.4% of patients reported a BADL improvement ≥ 2 points in the two periods, respectively. Decannulation rate and ability to walk at discharge did not change between the first and second 5-year period analyzed. Nearly 15% of patients who reached the goal to walk were able to do it autonomously, and this rate remained stable over both periods. Finally, the setting of discharge at the end of rehabilitation did not change between the two 5-year periods. However, days spent in the Weaning Unit were 55.4 ± 31.5 days in the period 2010–2014 and 45.1 ± 22.2 days in the period 2015–2019 (*p* = 0.01) (data not displayed).

### 3.3. Prediction of Rehabilitation Outcomes

Table 2 reports the correlation between patient characteristics at baseline with the four most relevant outcomes of rehabilitation recorded at T1 (weaning, decannulation, ability to walk, any BADL improvement) as dependent variables. The patients’ age was significantly associated with BADL improvement (*p* < 0.001) and to walking ability (*p* < 0.001), and borderline associations were found with weaning and decannulation. Sex was significantly associated only with BADL improvement (*p* = 0.03). The level of comorbidity was significantly associated with all four dependent variables, and the clinical severity was significantly associated with decannulation (*p* = 0.028) and BADL improvement (*p* = 0.008). Among diagnoses at admission, acute brain damage was negatively associated with BADL improvement and ability to walk (OR = 0.1 95%CI [0.03–0.7] *p* = 0.02 and OR = 0.2 95%CI [0.03–0.9], *p* = 0.01), while no other significant associations were detected with the pre-existing conditions (Table 3).

## 4. Discussion

This retrospective study reports the extensive clinical experience with difficult-to-wean tracheostomized patients referred to a specialized unit of a rehabilitation center. The patients’ anthropometry and preadmission clinical management in acute care hospitals were similar over time, although year by year, the categories of underlying disease changed (COPD decreased and other conditions increased), and the patients’ complexity increased. Despite this, the main treatment outcomes did not worsen, with the exception of a reduction in the proportion of patients achieving a BADL improvement of ≥ 2 points, which became evident in the comparison between the two 5-year periods considered. Both the patients’ level of comorbidities and clinical severity at admission were significant predictors of the clinical outcomes.

As a preliminary consideration, it must be noted that fewer than 60% of all tracheostomized patients admitted to our weaning unit participated in a comprehensive rehabilitation course to achieve the weaning from mechanical ventilation and to improve physical performance. There is consensus that a dedicated clinical setting for patients with difficult or prolonged weaning could help to identify those individuals at greater risk of failure and/or subsequent death [10,11]. In a population of 240 subjects recovering from acute respiratory failure and subjected to a comprehensive rehabilitation program, Vitacca et al. reported a weaning rate of 47%, a mortality rate of 13.8%, and a relative decannulation rate among survivors of 32%, thus comparable to what was reported in our cohort [12]. Chiang et al. showed that a 6-week physical training program improved the ventilator-free time in 20 difficult-to-wean patients admitted to a dedicated weaning unit [13]. These results were further confirmed in two subsequent randomized clinical trials conducted in 2011 and in 2012 [14,15].

To the best of our knowledge, Polverino et al. reported the largest-yet experience, with outcomes of over 3000 patients admitted to five specialized weaning units over 15 years [16]. In that study, however, the authors did not refer specifically only to those individuals who were long-tracheostomized at admission, a subgroup of patients who are difficult-to-wean by definition. Moreover, they did not state whether the patients recruited were able to participate in comprehensive rehabilitation, which should include peripheral muscle reconditioning. Therefore, the data obtained in our single center over a long period of time reinforce the evidence that intensive physiotherapy, which includes both respiratory exercises and progressive peripheral muscle reconditioning, is feasible for a proportion of tracheostomized patients. Indeed, several reasons, such as clinical instability and the inability to cooperate, often preclude some patients from participating (Figure 1).

The level of comorbidity of the study cohort changed over the years, reflecting the increasing complexity of these populations and the multiple possible causes of difficult weaning. It is very likely that the staff’s skills improved over time [17], thus allowing them to care for more complex patients in the intensive care unit. As a result, patients could have been treated earlier and received a more comprehensive approach to weaning [18], thereby greatly reducing the burden of prolonged care. Indeed, our retrospective analysis found that the length of stay in the weaning unit and the time to home discharge reduced slightly, although not significantly, over the entire 10-year period (data not shown).

Interestingly, the main clinical outcomes (i.e., weaning and/or decannulation success) achieved in this complex population of tracheostomized patients did not substantially change when comparing the two consecutive 5-year periods examined (Table 1, Part 2). We compared 5-year periods for two main reasons: first, because of the more homogeneous care approach in each period, and second, the differing organization and availability of beds in 2010–2014 compared to that in 2015–2019. The observed trend of fewer patients undergoing successful decannulation may be linked to the different diagnostic categories reported in the second period, when complex cardiac diseases, polytrauma, and neurological conditions were more represented (Table 1, part 1), resulting in a population less likely to be weaned [19]. On the other hand, weaning achievement tended to be better in the second period. It is possible that this may have been due to the smaller proportion of patients with COPD, who are often recognized as difficult-to-wean once tracheostomized [20], with an urgent need for advanced care [21].

Interestingly, a larger proportion of patients in the second 5-year period reported ability to walk at discharge from the Weaning Unit (Table 1, Part 2), despite the same period registered a lower proportion of patients who improved their BADL score ≥ 2 points during rehabilitation. In a previous study, it was shown that the mortality rate and weaning success differ according to BADL score following active rehabilitation/training in tracheostomized, ventilated, difficult-to-wean patients [22].

The tracheostomized individuals included in our weaning program were elderly patients with pre-existing chronic conditions. It is well known that the aging process is associated with motor and cognitive function decline, which leads to a reduction in autonomy in daily living activities, especially in patients who suffer from multiple comorbidities [23]. Again, the different diagnostic case mix could have influenced this result. Nonetheless, ability to walk is just one of the several indicators of response to rehabilitation in the critical care area [24]. Selective disability and reduced capacity to perform daily activities are meaningful to these individuals, especially when they cannot achieve decannulation. Therefore, given the changes in the population and the trend observed in the other main clinical outcomes, it is not surprising that BADL scores decreased in our study cohort over time.

Finally, our study confirmed that, even in difficult-to-wean tracheostomized patients, the outcomes of rehabilitation were best predicted by the patient’s age, comorbidities level, and clinical severity at admission (Table 2), as previously reported in the literature [25,26]. In particular, this is the first study to confirm that global complexity, as reflected by the level of comorbidities, is the most important negative predictor of clinical outcomes in difficult-to-wean tracheostomized patients referred to rehabilitation. Therefore, comorbidity should always be assessed in these patients at baseline. The CIRS was originally created to assess patient complexity in the internal medicine setting, and it has since been applied to different types of patients, such as those receiving critical care. This scale confirms its clinical validity in measuring multimorbidity [27], and it can be useful in predicting patient outcomes, thus facilitating the adoption of adequate qualitative and quantitative approaches, according to the resources available [28].

Overall, it appears that advanced age and higher CIRS and SAPS II scores in these patients are associated with worse recovery of autonomy in daily activities. Therefore, our results show that chronic damage affects recovery more than do the other variables; age and comorbidities are likely to compromise the patient’s ability to sustain exercise and the incremental muscle workload required during rehabilitation. It may be worth discussing if more compromised patients could benefit from a more intense rehabilitation program. Notwithstanding, cerebral ischemia was the only condition, among all those pre-existing in the population of patients admitted and treated, which negatively influenced outcomes (improvement in BADL and ability to walk) related to the physical function (see in Table 3). Therefore, it seems that patient complexity—more than each single condition—may affect the chance for a positive outcome following the rehabilitation course. Indeed, Martin et al. showed that patients receiving chronic ventilation are weak and deconditioned, but they may even respond to aggressive whole-body and respiratory muscle training associated with improvements in strength, weaning outcome, and functional status [29]. Large trials with appropriate design are required to investigate this issue.

This study has several limitations. First of all, as a retrospective analysis, it provides information that will have to be confirmed prospectively in the specific population studied. Second, we cannot exclude that the outcome assessments might have been influenced not only by the staff’s improved ability [17], but also by external factors such as resource availability and changes in the local healthcare system [16], which were not taken into account. Third, we did not explore the Functional Independence Measure (FIM) as a potential outcome [30]. Further, SAPS II was developed and validated in the critical care setting but not specifically in the population enrolled in the study (post-acute patients); thus, caution should be used when considering this score in this subset of patients [31]. Although other multi-factorial scores have been recently developed to better assess the physical morbidity of the general adult critical care population [32,33], the good performance of SAPS II in the intermediate care setting and its wide usage in the available literature [34] motivated our choice in assessing clinical severity of our population. Last, this study reflects the experience of a single regional weaning unit, and its results cannot be easily generalized to facilities with a different organization.

## 5. Conclusions

This study confirms and expands on the literature regarding the role of weaning units’ implementing comprehensive rehabilitation programs for the very severe, disabled subset of tracheostomized patients, and it shows, for the first time, a correlation between CIRS score and clinical outcomes at discharge.

These findings could therefore spur new research exploring the impact of comprehensive rehabilitation, including intensive physiotherapy in difficult-to-wean tracheostomized patients. Specifically, properly sampled randomized control trials aimed at figuring out tailored rehabilitation programs focused on this particular type of patient seem an urgent need.

## Figures and Tables

**Figure 1 ijerph-19-02982-f001:**
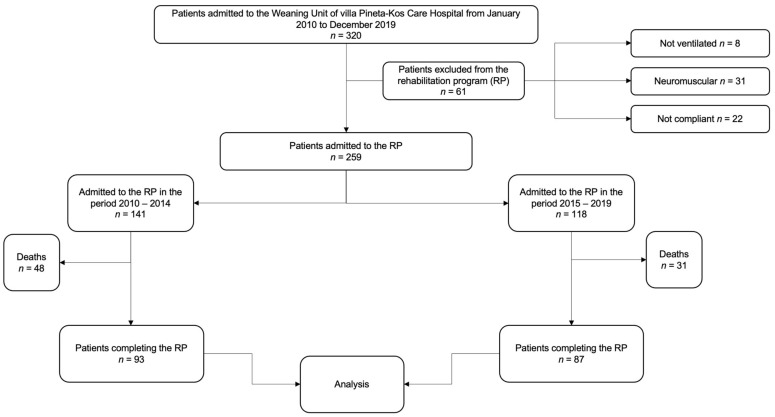
Study flowchart.

**Table 1 ijerph-19-02982-t001:** Baseline features and clinical outcomes of the study population. Part 1. Anthropometric data, diagnosis, and clinical severity at admission to the Weaning Unit, according to the two consecutive 5-year time periods and overall. Part 2. Rehabilitation outcomes for the overall population and according to the two consecutive 5-year time periods.

	Total	2010–2014	2015–2019	*p*-Value
Admitted to rehabilitation, *n* (%)	259 (100)	141 (54)	118 (46)	
Death, *n* (%)	79 (31)	48 (34)	31 (26)	0.18
Included in analysis, *n* (%)	180 (69)	93 (66)	87 (74)	0.22
* **Part 1. Baseline variables *** *				
Age, years (SD)	73 (10.1)	73.1 (9.9)	72.9 (10.2)	0.88
Male sex (%)	102 (56.7)	45 (48.4)	57 (65.5)	0.02
Baseline CIRS, score (SD)	19 (3.9)	18.2 (4.2)	20.0 (3.4)	0.002
Baseline SAPS II, score (SD)	21.5 (14.1)	19.7 (11.4)	23.4 (16.4)	0.08
Time in an acute care hospital, days (SD)	36.4 (26.7)	36.6 (25.0)	36.1 (28.6)	0.9
Time from tracheostomy, days (SD)	22.1 (22.3)	22.4 (20.8)	21.7 (23.9)	0.84
Diagnosis of admission:				
COPD, *n* (%)	75 (42)	41 (44)	34 (39)	0.54
Heart surgery, *n* (%)	32 (18)	20 (22)	12 (14)	0.24
Abdominal surgery, *n* (%)	22 (12)	16 (17)	6 (7)	0.04
Polytrauma, *n* (%)	20 (11)	7 (8)	13 (15)	0.15
Metabolic syndrome, *n* (%)	9 (5)	2 (2)	7 (8)	0.09
Thoracic surgery, *n* (%)	6 (3)	2 (2)	4 (5)	0.43
Acute brain damage, *n* (%) (hemorrhage/ischemia)	9 (2)	3 (2)	6 (3)	0.19
OSAS, *n* (%)	4 (2)	1 (1)	3 (3)	0.35
Pulmonary fibrosis, *n* (%)	2 (1)	1 (1)	1 (1)	0.9
WNV encephalitis, *n* (%)	1 (0.6)	0 (0)	1 (1)	0.48
* **Part 2. Clinical outcomes *** *				
Successful weaning, *n* (%)	119 (66.1)	55 (59.1)	73.6/26.4	0.06
Decannulation, *n* (%)	82 (45.6)	47 (50.5)	40.2/59.8	0.17
Ability to walk, *n* (%)	106 (58.9)	51 (54.8)	63.2/36.8	0.26
BALD change				
*≤0, n (%)*	48 (26.7)	18 (19.4)	30 (34.5)	0.03
*+1, n (%)*	42 (23.3)	21 (22.6)	21 (24.1)	0.86
*+2, n (%)*	34 (18.9)	19 (20.4)	15 (17.3)	0.7
*+3, n (%)*	20 (11.1)	15 (16.1)	5 (5.7)	0.03
*+4, n (%)*	9 (5)	4 (4.3)	5 (5.7)	0.74
*+5, n (%)*	10 (5.6)	6 (6.5)	4 (4.6)	0.75
*+6, n (%)*	17 (9.4)	10 (10.8)	7 (8.1)	0.8
Discharged, *n* (%)	53 (29.4)	27 (29)	26 (29.9)	0.9
Transfer to a residential care facility, *n* (%)	97 (53.9)	47 (50.5)	50 (57.5)	0.37
Transfer to acute care hospital, *n* (%)	30 (16.7)	19 (20.4)	11 (12.6)	0.23

Legend: SD = standard deviation; M = male; COPD = chronic obstructive pulmonary disease; OSAS = obstructive sleep apnea syndrome; WNV = West Nile virus; CIRS = cumulative illness rating scale; SAPS = simplified acute physiology score. BADL = Bristol Activities of Daily Living. * these data are referred to those patients included in the analysis plan.

**Table 2 ijerph-19-02982-t002:** Association between characteristics of patients at admission and clinical rehabilitation outcomes.

		AGE	SEX	CIRS	SAPS II %	Days at Acute Hospital	Timing of Tracheostomy
Weaning	R^2^_McF_	0.01	0.00165	0.0153	0.00721	0.00111	5.86 × 10^−4^
	*p*-value	0.05	0.48	**0.03**	0.14	0.57	0.68
	Association	No (borderline significant)	No	Yes	No	No	No
Decannulation	R^2^_McF_	0.0154	8.01 × 10^−5^	0.0497	0.0207	0.00369	0.00176
	*p*-value	0.05	0.89	<0.001	0.28	0.352	0.517
	Association	No (borderline significant)	No	Yes	Yes	No	No
Ability to walk	R^2^_McF_	0.0297	0.00592	0.0757	0.00434	0.00308	0.00334
	*p*-value	0.01	0.23	< 0.001	0.3	0.39	0.38
	Association	Yes	No	Yes	No	No	No
Any BADL improvement	R^2^_McF_	0.0305	0.00137	0.0896	0.0111	0.0021	1.36 × 10^−4^
	*p*-value	< 0.001	0.03	< 0.001	0.008	0.25	0.77
	Association	Yes	No	Yes	Yes	No	No

Legend: CIRS = Cumulative Illness Rating Scale; SAPS = Simplified Acute Physiology Score; BADL = Bristol Activities of Daily Living.

**Table 3 ijerph-19-02982-t003:** Raw association between underlying diagnosis at admission and the rehabilitation outcomes. Association is shown through odds ratio (OR) and 95%CI.

	Weaning			Decannulation			Ability to Walk			Any BADL Improvement		
Diagnosis	OR	95%CI	*p*-Value	OR	95%CI	*p*-Value	OR	95%CI	*p*-Value	OR	95%CI	*p*-Value
COPD	1.8	0.9–3.3	0.08	1.2	0.7–2.2	0.5	0.8	0.5–1.5	0.6	0.9	0.4–1.7	0.7
Heart surgery	1.3	0.6–3.2	0.5	1.1	0.5–2.3	0.9	0.6	0.3–1.4	0.3	0.8	0.3–1.8	0.5
Abdominal surgery	0.8	0.3–2.2	0.8	1.2	0.5–3	0.7	1	0.4–2.5	0.9	1.7	0.6–5.4	0.3
Polytrauma	0.9	0.4–2.5	0.9	1.9	0.7–5	0.2	1.3	0.5–3.5	0.7	1.1	0.4–3.2	0.9
Metabolic syndrome	4.4	0.7–50	0.2	0.9	0.2–3.7	0.9	1.4	0.3–5.9	0.6	1.3	0.3–6.4	0.8
Thoracic surgery	2.6	0.3–23	0.4	2.5	0.4–14	0.3	1.4	0.2–7.9	0.7	1.9	0.2–16	0.6
Acute brain damage (ischemia/hemorrhage)	1	0.3–4.3	0.9	0.3	0.1–1.6	0.1	0.2	0.03–0.9	0.02	0.1	0.03–0.7	0.01
OSAS	1.6	0.2–15	0.7	1.2	0.2–8.7	0.9	2.1	0.2–21	0.5	1	0.1–11	0.9
Pulmonary fibrosis	0.5	0.03–8.3.	0.6	0.2	0.01–4.9	0.2	3.6	0.2–75	0.2	1.9	0.1–39	0.4
WNV encephalitis	1.6	0.1–39	0.5	0.4	0.01–9.8	0.4	2.1	0.08–53	0.9	1.1	0.04–28	0.5

Legend: CI = Confidence Interval; COPD = chronic obstructive pulmonary disease; OSAS = obstructive sleep apnea syndrome; WNV = West Nile virus; OR = odds ratio; BADL = Bristol Activities of Daily Living.

## Data Availability

The data underlying this article cannot be shared publicly due to the privacy of individuals that contributed to the study with their data. The data can be shared on reasonable request to the corresponding author.

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
