# Peer review of "Rehabilitation of Difficult-to-Wean, Tracheostomized Patients Admitted to Specialized Unit: Retrospective Analyses over 10-Years"

_ijerph, 2022, doi:10.3390/ijerph19052982_

Round 1
Reviewer 1 Report
In this manuscript, it seems that the authors wanted to compare the effects of rehabilitation in post-tracheostomy patients according to the inclusion period.
The reviewers wondered if any of the authors were familiar with medical statistics. The study design, the method of analysis, and the way it is presented are all a mess. The reviewer hopes that the reviewers' comments can help improve the manuscript.
The timing of exclusion is quite important and should be precisely defined and presented because the present study is comparing groups divided based on the timeline. For example, “Deceased Patients” and “Unable to participate” have completely different meanings. The former patients were lost during the follow-up period, while the latter patients were excluded from the beginning. Therefore, the number of excluded or lost patients may change according to the studied period (before or after 2015), leading to selection bias.
Given the above problems, it is no surprise that Figure 1 is of poor quality. Good examples can be found elsewhere, but the following article is one of which: Yang et al. (2018) https://doi.org/10.1093/ejcts/ezx215. (PMID: 29106506). It is suggested that the finally analyzed population be placed in the center of the chart and that the compared two groups be shown as well. Remember that the first juncture in that flowchart is whether the observation period is before or after 2015.
For all tables, the "Total" column should be on the far left of the table, not to the right of the "p-value" column. Although the comparison is between two groups separated by time, there is no correspondence between the subjects of the two groups, so it is wrong to perform the Kruskal-Wallis test. For the categorical variables, only one p-value (0.098) is shown, which is difficult for the reviewer to understand why.
Author Response
REVIEWER 1
Comments and Suggestions for Authors
Reviewer 1’s general comment
In this manuscript, it seems that the authors wanted to compare the effects of rehabilitation in post-tracheostomy patients according to the inclusion period.
The reviewers wondered if any of the authors were familiar with medical statistics. The study design, the method of analysis, and the way it is presented are all a mess. The reviewer hopes that the reviewers' comments can help improve the manuscript.
We thank the Reviewer for the accurate reviewing process of our work. We have welcomed all of his/her comments and we have tried to emend the manuscript accordingly.
Reviewer 1’s comment 1
The timing of exclusion is quite important and should be precisely defined and presented because the present study is comparing groups divided based on the timeline. For example, “Deceased Patients” and “Unable to participate” have completely different meanings. The former patients were lost during the follow-up period, while the latter patients were excluded from the beginning. Therefore, the number of excluded or lost patients may change according to the studied period (before or after 2015), leading to selection bias.
Response to Reviewer 1’s comment 1
We thank the Reviewer for this comment. We agree with the Reviewer that this point represents a major flaw of our work. We have now redrawn Figure 1 in order to clarify the timing of exclusion and the proportion of patients who died during the rehabilitation course. We have specified this point in the methods section as follows: “All patients admitted to the Weaning Unit among two subsequent time periods (namely 2010-2014 and 2015-2019) were considered eligible for the analysis. Patients with degenerative neuromuscular diseases, those for whom the outcome of weaning and decannulation cannot be predicted beforehand and those who were considered not compliant to the rehabilitation program were excluded a priori. Patients who died during hospital stay were further excluded; thus, only those who completed the rehabilitation program were included in the analysis plan. Details on patient selection are shown in Figure 1”.
Reviewer 1’s comment 2
Given the above problems, it is no surprise that Figure 1 is of poor quality. Good examples can be found elsewhere, but the following article is one of which: Yang et al. (2018) https://doi.org/10.1093/ejcts/ezx215. (PMID: 29106506). It is suggested that the finally analyzed population be placed in the center of the chart and that the compared two groups be shown as well. Remember that the first juncture in that flowchart is whether the observation period is before or after 2015.
Response to Reviewer 1’s comment 2
We thank the Reviewer for this comment. We have now redrawn the Figure 1 according to his/her suggestion.
Reviewer 1’s comment 3
For all tables, the "Total" column should be on the far left of the table, not to the right of the "p-value" column.
Response to Reviewer 1’s comment 3
We thank the Reviewer for this comment. We have changed the Table accordingly.
Reviewer 1’s comment 4
Although the comparison is between two groups separated by time, there is no correspondence between the subjects of the two groups, so it is wrong to perform the Kruskal-Wallis test. Response to Reviewer 1’s comment 4
We thank the Reviewer for this comment. We do apologize for this mistake. We actually performed t test for continuous variables. We have now emended the manuscript.
Reviewer 1’s comment 5
For the categorical variables, only one p-value (0.098) is shown, which is difficult for the reviewer to understand why.
Response to Reviewer 1’s comment 5
We thank the Reviewer for this comment. We have now reported all the p values as appropriate.
Reviewer 2 Report
The present study involved a retrospective review of over 10 years of tracheostomized patients to determine rehabilitation impact on patients' recovery post-tracheostomy. The study adds valuable evidence to the literature on this important topic for clinicians. The scientific soundness of the study design is high; while it would have been nice to include more quantitative statistical analyses, the authors are limited to the number and nature of patients they had at their disposal for this analysis. I would suggest the authors more heavily advocate in their Discussion/Conclusions for more multi-center randomized controlled trials to determine specific rehabilitation protocols in this particular population of patients. This would add increased value to the evidence shown in this retrospective cohort study.
Author Response
REVIEWER 2
Comments and Suggestions for Authors
Reviewer 2’s general comment
The present study involved a retrospective review of over 10 years of tracheostomized patients to determine rehabilitation impact on patients' recovery post-tracheostomy. The study adds valuable evidence to the literature on this important topic for clinicians.
The scientific soundness of the study design is high; while it would have been nice to include more quantitative statistical analyses, the authors are limited to the number and nature of patients they had at their disposal for this analysis.
We thank the Reviewer for the accurate reviewing process and the appreciation of our work. We have welcomed all of his/her comments and we have tried to emend the manuscript accordingly.
Reviewer 2’s comment 1
I would suggest the authors more heavily advocate in their Discussion/Conclusions for more multi-center randomized controlled trials to determine specific rehabilitation protocols in this particular
Response to Reviewer 2’s comment 1
We thank the Reviewer for this comment. We have now stressed this point in the conclusion as suggested. We have added the following paragraph: “In particular, properly sampled randomized control trials aiming at figuring out tailored rehabilitation programs on this peculiar type of patients seem an urgent need”.
Round 2
Reviewer 1 Report
The revised manuscript has now reached a level that allows for a proper peer review. The reviewer points out first the minor problems:
- The flowchart shown in Figure 1 was remarkably improved. Now we see that the percentage of deaths in each group is 34.0% vs 26.3% (2010-2014 vs 2015-2019, p=0.176). The reviewer would suggest adding this information to Table 1 or somewhere else.
- The layout of Table 1 has been also improved. One thing left undone is to show not only the absolute number of categorical variables but its percentages.
- Table 2 also requires a similar correction. The “Y/N ratio %” is a too redundant way for data presentation. The “absolute value with percentage” is the best traditional way. A research article, recently published on BMJ Open (PMID: 34980623), which Dr. Enrico Clini co-authors, will be a quite good example to show Tables.
Now, we can move on to the major issues:
The most important purpose of this study should be to identify the characteristics of “difficult-to-wean tracheostomized patients”. Therefore, a univariate analysis should be done between “decannulated or weaned patients” and “patients who failed to be weaned or decannulated”. Sub-group analysis according to the study period (2010-2014 vs 2015-2019) has little importance. In other words, Table 3 is much more important than Table 1 or Table 2.
Author Response
REVIEWER 1
Comments and Suggestions for Authors
Reviewer 1’s general comment
The revised manuscript has now reached a level that allows for a proper peer review.
We thank the Reviewer for his/her comment and for the further effort to review our manuscript R1.
Reviewer 1’s Minor problems
Comment 1
The flowchart shown in Figure 1 was remarkably improved. Now we see that the percentage of deaths in each group is 34.0% vs 26.3% (2010-2014 vs 2015-2019, p=0.176). The reviewer would suggest adding this information to Table 1 or somewhere else.
The layout of Table 1 has been also improved. One thing left undone is to show not only the absolute number of categorical variables but its percentages.
Response to comment 1
We thank the Reviewer for his/her comment. We agree that Table 1 could be misleading and redundant. We have therefore decided to change it according to suggestions and we have merged data with those in Table 2, which are now visible in the uploaded manuscript version R2.
Comment 2
Table 2 also requires a similar correction. The “Y/N ratio %” is a too redundant way for data presentation. The “absolute value with percentage” is the best traditional way. A research article, recently published on BMJ Open (PMID: 34980623), which Dr. Enrico Clini co-authors, will be a quite good example to show Tables.
Response to comment 2
We thank the Reviewer for this comment. Data in Table 2 have been merged with those of Table 1 into one new Table: see also reply to minor comment 1.
Reviewer 1’s Major issues
Comment 1
The most important purpose of this study should be to identify the characteristics of “difficult-to-wean tracheostomized patients”. Therefore, a univariate analysis should be done between “decannulated or weaned patients” and “patients who failed to be weaned or decannulated”. Sub-group analysis according to the study period (2010-2014 vs 2015-2019) has little importance. In other words, Table 3 is much more important than Table 1 or Table 2.
Response to comment 1
We thank the Reviewer for this comment, and we agree with the point. Therefore, we have now added as supplementary Table the univariate univariable analysis considering the YES/NO risk to have each of the 4 considered outcomes following comprehensive rehabilitation (see Methods). Also, we have modified Discussion section accordingly. Different analysis on composite outcomes (i.e. weaned+decannulated, or so on) have not been considered further not to further under-power the relationship which is per se low due to the study characteristics (retrospective analysis, monocentric design, no sample matching a priori, etc.).

Round 3
Reviewer 1 Report
Nothing changed from ver.2
Where is the author's response letter?
Author Response
Please see appended supplementary Table